# Cognitive Decline in Nasopharyngeal Carcinoma Survivors with Post-Radiation Epilepsy: A Prospective Cohort Study

**DOI:** 10.3390/cancers17121976

**Published:** 2025-06-13

**Authors:** Kejia Liu, Yaxuan Pi, Yingying Zhu, Dong Pan, Zongwei Yue, Yanting Chen, Lianhong Yang, Yituan Xie, Yuhua Huang, Yamei Tang, Yongteng Xu, Xiaoming Rong

**Affiliations:** 1Department of Neurology, Sun Yat-sen Memorial Hospital, Sun Yat-sen University, Guangzhou 510120, China; liukj5@mail2.sysu.edu.cn (K.L.); piyx@mail2.sysu.edu.cn (Y.P.); zhuyy79@mail.sysu.edu.cn (Y.Z.); pand6@mail2.sysu.edu.cn (D.P.); yuezw@mail.sysu.edu.cn (Z.Y.); chenyt367@mail.sysu.edu.cn (Y.C.); docylh@163.com (L.Y.); tangym@mail.sysu.edu.cn (Y.T.); 2Department of Neurology, Huizhou First Hospital, Huizhou 516003, China; xyt39410@163.com; 3Department of Neurology, Chaozhou People’s Hospital, Chaozhou 521011, China; hyh730420@sina.com

**Keywords:** cognitive function, epilepsy, nasopharyngeal carcinoma, radiotherapy

## Abstract

Epilepsy is a common complication after radiotherapy and may lead to accelerated cognitive ageing. However, it is unknown whether and how much faster NPC patients with epilepsy experience cognitive decline beyond the expected radiotherapy-related cognitive change. Here, we utilized clinical data from a prospective, registry-based cohort study to compare the rates of cognitive decline among NPC survivors with or without epilepsy. Our results found that Global cognitive function declined more rapidly in NPC patients with prevalent epilepsy compared with those without epilepsy. It indicated that early identification and control of seizure attack is extremely valuable to mitigate cognitive decline.

## 1. Introduction

Radiotherapy (RT) is an important treatment for head and neck cancers, especially for nasopharyngeal carcinoma (NPC) [1]. However, radiation may lead to some irreversible neurological complications [2], especially as the survival rate after tumor treatment increases. Epileptic seizures are one of the late symptomatic complications after RT and are associated with significant morbidity. Previous studies reported that recurrent seizures, epileptiform discharges, and the types and number of anti-seizure medications (ASMs) in use could lead to cognitive impairment [3,4,5]. Thus, in addition to controlling seizure attacks, cognitive decline is also a major concern for NPC survivors with epilepsy, which could significantly affect the quality of life of these patients.

In a previous study, we described the annual decline rate of cognition in NPC patients with radiation-induced brain necrosis (RN) [6]. However, we did not focus on the variance in cognitive function among patients with different comorbidities. For NPC survivors with epilepsy, their cognitive impairment is attributable to a combination of risk factors, including age, the treatment strategy for cancer, RN, recurrent seizure attacks, and vascular risk factors (VRFs, such as hypertension, diabetes, hyperlipidemia, and a history of cerebrovascular disease) [6,7,8,9]. It is unknown whether and how much faster NPC patients with epilepsy experience cognitive decline beyond the expected radiotherapy-related cognitive change. Furthermore, the association of some modifiable factors such as VRFs and serum inflammation level with cognitive decline have been proven in patients with epilepsy [8,10,11]. Whether this association still exists in NPC survivors with epilepsy has not been addressed. Thus, there is a pressing need to elucidate the cognitive trajectory and modifiable factors that hasten cognitive decline in these patients, as it would help to guide the development of interventions for preserving cognitive health in this population.

To address this question, we conducted a prospective, registry-based cohort study to compare the rates of cognitive decline among NPC survivors with and without epilepsy. Considering that different kinds of head and neck tumors receive different radiation strategies, we only included patients with NPC to correct for this confounding factor. We hypothesized that NPC survivors with active epilepsy experience accelerated cognitive deterioration compared to those without epilepsy.

## 2. Methods

### 2.1. Patient Cohort and Eligibility

This study was based on a prospectively ongoing cohort study, recruiting head and neck adult cancer patients with radiotherapy-related nervous system complications, conducted at Sun Yat-Sen Memorial Hospital, Sun Yat-Sen University, Guangzhou, China (NCT03908502). In our cohort, cognition evaluations were performed every 6 months. Between January 2005 and December 2023, 806 consecutive NPC patients with a history of radiotherapy who had undergone a baseline cognition assessment were screened. Data were extracted from electronic medical records. Patients were excluded if they met the following criteria: (1) their baseline major clinical data were unobtainable; (2) they displayed intracranial brain metastasis during follow-up; (3) they received a lack of follow-up on cognitive function for more than six months; and (4) they had a confirmed diagnosis of epilepsy before radiotherapy.

This study was approved by the Ethics Committee of Sun Yat-sen Memorial Hospital, Sun Yat-sen University (SYSEC-KY-KS-014, 14 June 2017), and all patients signed an informed consent form.

### 2.2. Baseline Data Collection

The baseline was defined as the date of patients receiving the first cognition assessment. Detailed baseline information was obtained from patients’ medical records, including demographic data (date of birth, gender, education, smoking and drinking history), prior tumor-related characteristics (TNM stage according to 7th edition of the AJCC/UICC staging system, RT techniques, radiation dose, and chemotherapy), medical history (hypertension, diabetes mellitus, stroke, radiation-induced brain necrosis, hypothyroidism, and extracranial arterial stenosis), laboratory evaluations, epilepsy parameters (latency to seizure and status), and ASM details (use of ASM or not and type of drugs).

### 2.3. Exposure Variable

The diagnosis of epilepsy was established by two neurologists based on the diagnosis criteria of the International League against Epilepsy 2017 guidelines [12,13]. Active epilepsy means having experienced at least one unprovoked seizure in the preceding 12 months, regardless of whether they were taking ASM or not. To investigate the impact of seizure control on cognitive function, we identified seizures as a major variable. We defined incident epilepsy cases as a group of patients whose frequency of seizure attacks was less than twice per year after enrollment into our study. The remaining epilepsy cases were regarded as prevalent epilepsy cases. Participants with no epilepsy were set as a comparison group.

### 2.4. Outcome Variables

The outcome was cognitive function assessed by the Chinese version of the Montreal Cognitive Assessment (MoCA) [14], with good sensitivity and specificity [15]. The MoCA questionnaire is a 30-point scale which consists of seven sub-items pertaining to visual function (score 0–5), naming (0–3), attention (0–6), language (0–3), abstraction (0–2), memory (0–5), and direction function (0–6). Lower scores indicate more severe cognition dysfunction. The cutoff point is 25/26 and with 1-point correction for people educated for no more than 12 years. Assessments of cognition were performed every 6 months through face-to-face interviews by trained interviewers who were certified in cognition assessment and were blind to the seizure status.

### 2.5. Covariates

Covariates included age, gender, body mass index, education, smoking, drinking, TNM stage, radiotherapy (RT) techniques (intensity-modulated radiation therapy or not), radiation dose, chemotherapy, comorbidities present at baseline, epileptic details, neutrophil-to-lymphocyte ratio (NLR), low-density lipoprotein (LDL) cholesterol level, high-density lipoprotein (HDL) cholesterol level, c-reactive protein (CRP), and erythrocyte sedimentation rate (ESR). Comorbidities included hypertension, diabetes mellitus, stroke, RN, hypothyroidism, and extracranial arterial stenosis. Epileptic details included onset of seizure after radiotherapy (years), the presence/absence of status epilepticus, and the number of ASMs in use.

### 2.6. Statistical Analysis

Baseline categorical variables were reported as the number of cases and percentages. Continuous variables conforming to normal distribution were expressed by the means and standard deviations, while those not conforming to normal distribution were described by the median (interquartile range, IQR). Univariate comparisons of baseline characteristics among the three groups were performed using χ^2^ tests, Mann–Whitney tests, or ANOVA tests according to the type and distribution of variables.

We used repeated measures of cognition and linear mixed models to assess the associations of prevalent and incident epilepsy with average cognitive trajectories during follow-up. The linear mixed model for MoCA score was as follows:

E (cognitive score) = intercept + epilepsy status + times + baseline cognition score + adjustment covariates + random effects included patients and times.

Variables with a *p* value < 0.1 in univariate analysis or of clinical interest (age, gender, education, intensity-modulated radiation therapy, radiation dose, hypertension, diabetes mellitus, stroke, usage of ASM, RN, NLR and CRP) [8,11,12] were further evaluated in the multivariable linear mixed model.

In order to explore whether the effect of epilepsy status on cognitive trajectories varied with different covariates, subgroup analyses was conducted according to age, gender, comorbidities, RT strategy, chemotherapy, RN, usage of ASM, NLR, and CRP. The interaction *p* values were calculated between epilepsy status and the above covariates.

All reported *p* values were 2-sided, with the level of significance defined as *p* < 0.05. Statistical analyses were performed in the R software for Windows (Version 4.1.3, R Core Team).

## 3. Results

Among the 806 NPC patients with cognitive assessment screening from January 2005 to December 2023, 538 patients underwent at least a 6-month cognition follow-up. We further excluded 17 patients lacked baseline data. After the application of our inclusion and exclusion criteria, 521 patients were recruited to our cohort, of whom 67 patients had epilepsy (42 incident epilepsy and 25 prevalent epilepsy) and 454 patients did not. Of the 67 patients with epilepsy, twenty-two had focal epilepsy and forty-five had generalized epilepsy. The median latency from radiotherapy to the first seizure attack was 7.33 years. A detailed flowchart of study cohort selection and grouping is given in Figure 1.

### 3.1. Baseline Patient Characteristics

Baseline patient characteristics are summarized in Table 1. The median age at study entry was 49.9 years (IQR, [26.1, 73.4]) and 26.5% (138/521) of participants were women. More patients in the prevalent epilepsy group suffered from diabetes and RN at study entry than those with incident epilepsy or non-epilepsy. Patients with prevalent epilepsy also had a higher CRP level compared with those with incident epilepsy or non-epilepsy (median CRP [IQR], 2.21 [0.81, 5.97] vs. 1.89 [1.14, 5.81] vs. 2.57 [0.51, 5.63], *p* = 0.036). Overall, 10.8% patients in the no epilepsy group had hypothyroidism, which was much higher than the percentage of those in epilepsy group (*p* = 0.021). The education level among the three groups was also significantly different (*p* = 0.028).

Of the 67 patients with epilepsy, only forty-nine were taking ASM at baseline, while the remaining eighteen had been newly diagnosed with epilepsy. Fifteen patients (15 of 67, 22.39%) took more than one drug. Only one patient complained of having status epilepticus.

As shown in Table 2, at baseline cognitive assessment, the mean MoCA total scores in the three groups were 22.85 (5.00), 22.11 (5.44), and 24.04 (4.04), respectively (*p* = 0.793). Overall, 64.10% of patients without epilepsy were recognized as displaying cognitive dysfunction at baseline, while 64.29% and 68.0% of patients with incident or prevalent epilepsy were recognized as displaying cognitive dysfunction (*p* = 0.525). For the seven sub-items of MoCA, patients without epilepsy seemed to have a lower score in naming compared with those in the epilepsy groups (*p* = 0.018).

### 3.2. Cognitive Function Changes During Follow-Up

We estimated longitudinal MoCA changes over a median follow-up period of 3.96 years (IQR 2.0 to 7.74 years). The total score and the seven sub-items scores gradually decreased during follow-up (Figure 2). As shown in Table 3, the rate of decline was significantly faster in the prevalent epilepsy group compared with the no epilepsy group (*p* = 0.007) after adjusting for demographics, health behaviors, tumor-related history, radiation dose, complications, use of ASM, and inflammatory blood index. However, the cognitive decline rate was similar in the incident epilepsy group compared with that in the non-epilepsy group (*p* = 0.126). Moreover, multivariable analysis showed that, in addition to epilepsy status, age, education level, hypertension, measurement times (reflecting the follow-up period), and baseline MoCA scores were also significantly associated with cognitive decline (Table 4).

For the seven sub-items, although attention, language, abstract, memory, and direction scores seem to be significantly decreased in the prevalent epilepsy group compared with those in the no epilepsy group, in multivariable analysis, only abstract, memory, and direction scores remained significantly different during follow-up as compared with those in the no epilepsy group (abstract score: Est, −0.224, 95%CI −0.408, −0.041, *p* = 0.016; memory score: Est −0.581, 95%CI −0.969, −0.194, *p* = 0.003; direction score: Est −0.243, 95%CI −0.453, −0.032, *p* = 0.024) (Table 3).

### 3.3. Subgroup Analysis

Considering that age, gender, vascular risk factors, radiation strategy, chemotherapy, RN, systematic inflammation, and usage of ASM have all been reported to be related to cognition, we further explored whether the effects of epilepsy status varied in the subgroups defined according to the above covariates, and the interaction *p* values were calculated. However, as shown in Figure 3, there was no significant difference in the effect of epilepsy status on cognitive deterioration among subgroups stratified by the pre-planned covariates.

## 4. Discussion

To the best of our knowledge, this is the first registry-based cohort study to investigate cognitive function in NPC patients combined with epilepsy after radiotherapy. We found that at baseline, over 60% of the patients suffered from mild cognitive impairment, although most of them were admitted and recruited to our cohort due to radiation-related complications (such as RN and radiation-related cranial nerve injury) rather than cognitive dysfunction. The baseline cognitive function assessed by MoCA was similar among patients with and without epilepsy. However, over a median follow-up period of 3.96 years, the rate of cognitive decline was significantly faster in the prevalent epilepsy group compared with the no epilepsy group after adjusting for the potential confounding factors. Noticeably, the change in cognitive function was similar between those with good epilepsy control and those without epilepsy.

In recent years, there has been growing evidence that patients with epilepsy are likely to develop significant cognitive dysfunction. Common cognitive deficits in people with epilepsy include reduced information processing speed, memory impairments, intellectual decline, and attentional deficits [16,17,18]. The most vulnerable domain of cognitive impairment is related to the location of epilepsy. Memory is the most frequently reported problem in patients with temporal lobe epilepsy (TLE) and medial temporal lobe epilepsy. Bender’s study included a total of 81 adult patients with TLE and 28 normal comparisons with similar mean ages. Worse memory performance and diminished executive function were observed in TLE patients compared with the normal comparisons [19]. Language, attention impairment, executive dysfunction, and cognitive control problems such as inhibition and shifting dysfunction were reported to be correlated with frontal lobe epilepsy [20,21,22], while patients with parietal and occipital lobe epilepsy performed significantly worse in visuo-construction, as well as verbal and executive functions [23,24]. For NPC patients, RN most frequently occurs in the temporal lobe after radiotherapy, and the surrounding area of the RN lesion is prone to triggering an epileptic zone [25,26]. In our study, the mean baseline score of the memory domain was relatively low, which may be related to the radiation injury in the temporal lobes. During the follow-up, the cognitive function in all the three groups showed a gradual decline trend, and memory deficit was found to be the most prominent manifestation of cognitive impairment in NPC patients with epilepsy compared with those without epilepsy, which is consistent with the pathological changes in this population.

Even with the same lesion, cognitive dysfunction is also mediated by the focus side, the localization of the seizure zone, as well as the propagation pattern of seizures and their frequency [27,28]. In addition, the patients’ situation in our cohort is particularly complex because apart from epilepsy, the treatment for NPC (radiotherapy, chemotherapy), the RN lesion, old age, the combined vascular risk factors (hypertension, diabetes, history of stroke), epilepsy details (seizure frequency, seizure types), as well as the anti-seizure medication can impact cognitive functioning. Thus, it is difficult to identify which is the most decisive factor for cognitive decline in this population. In our analysis, we set epilepsy frequency as the exposure variable and used multivariable linear mixed model to address the possible independent risk factors for cognitive deterioration. Although the rate of cognitive decline was significantly faster in the prevalent epilepsy group compared with the no epilepsy group, no significant difference was observed in the incident epilepsy group compared with the no epilepsy group. It seems that cognitive impairment in NPC patients arising from seizure activity can be reduced or reversed by effective seizure control, which is in line with previous research [29]. Thus, the timely identification and early treatment of seizure attacks to achieve a freedom from seizures in patients after radiotherapy is extremely valuable for avoiding faster cognitive decline than expected.

Growing evidence has shown that vascular risk factors and comorbid cardiovascular disease have been linked to faster cognitive decline in older adults than expected [8,30,31,32]. Radiotherapy for head and neck cancers can cause vascular complications and metabolic disorder [2,33,34,35]. Due to the limitation of sample size and data accessibility, we could not include all the vascular and metabolic risk factors in the multivariable analysis, but we still try to include as many relevant variables as possible. Our results demonstrate that in addition to epilepsy status, age, education level, hypertension, and baseline MoCA score were significantly associated with cognitive decline. We speculated that vascular risk factors might affect the integrity of brain networks, which could aggravate the deterioration of cognitive function. In the multivariate model, we did not assess the independent effect of ASM on cognitive outcome. We speculate that this may be due to the fact that most patients use second-generation ASMs, such as oxcarbazepine, sodium valproate, and levetiracetam, which are rarely reported to cause cognitive impairment.

In subgroup analysis, there was no significant difference in the effect of epilepsy status on cognitive deterioration among subgroups stratified by the pre-planned covariates. Considering that the confidence intervals of some subgroups are large, further analysis with larger sample size may be needed to validate our findings.

There are several important limitations of our study. First, although it is as prospective study, some patients may have been lost to follow-up because of disease progression or unexpected death, which could lead to potential biases. Second, some important information on radiotherapy, for example, radiation fractionation, was not available. In order to minimize the confounding factors of radiation strategy, we only included patients with NPC. In addition, we did not have data on brain neuroimaging and EEG parameters at baseline, as well as other psychiatric comorbidities (such as depression and anxiety) which are commonly seen in patients with malignancy and can also impact cognitive functions. Third, some of our participants had concurrent RN and may have received corticosteroid or bevacizumab therapy [36,37]. The changing volume of RN during the follow-up period might also have partially affected the cognition evaluation. Fourth, we used data collected from 2005 to 2023 in order to enroll as many patients as possible. Despite the cohort being large, the number of patients with epilepsy was still small (only 67 patients), likely limiting the power of the analysis. Fifth, MoCA is only a screening tool for cognitive dysfunction. The diagnosis of dementia and determination of its severity involve a combination of patient medical history, physical signs, imaging examinations, and scale assessments. In the future, using more elaborative tools may help to evaluate the cognitive function better. Finally, changes in epilepsy treatment (drugs and surgery), seizure frequency, and RN treatment (corticosteroid and bevacizumab) strategies during follow-up may also have important implications for cognitive health, which have not been fully evaluated in our study.

## 5. Conclusions

Our study showed that cognitive dysfunction is a common problem in NPC patients with epilepsy. Furthermore, global cognitive function declined more rapidly in NPC patients with prevalent epilepsy compared with those without epilepsy. Reducing the frequency of seizure attacks and treating hypertension may represent modifiable factors in slowing cognitive decline. Further studies seeking to find the biological mechanisms explaining this association and potentially identifying further modifiable risk factors for slowing down cognitive deterioration are warranted.

## Figures and Tables

**Figure 1 cancers-17-01976-f001:**
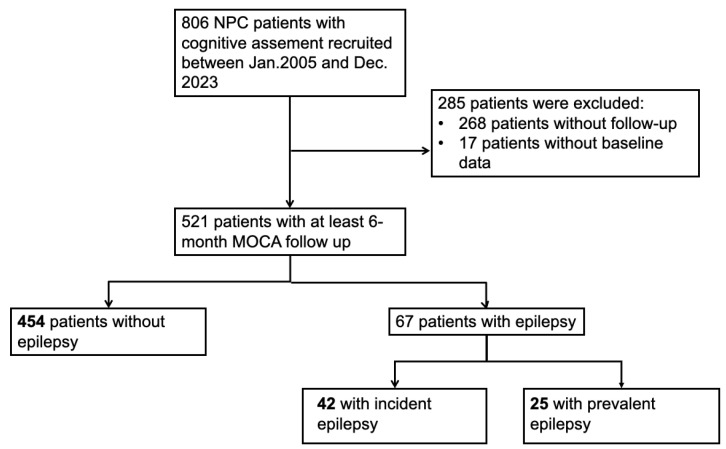
Flowchart of selecting patients in this study.

**Figure 2 cancers-17-01976-f002:**
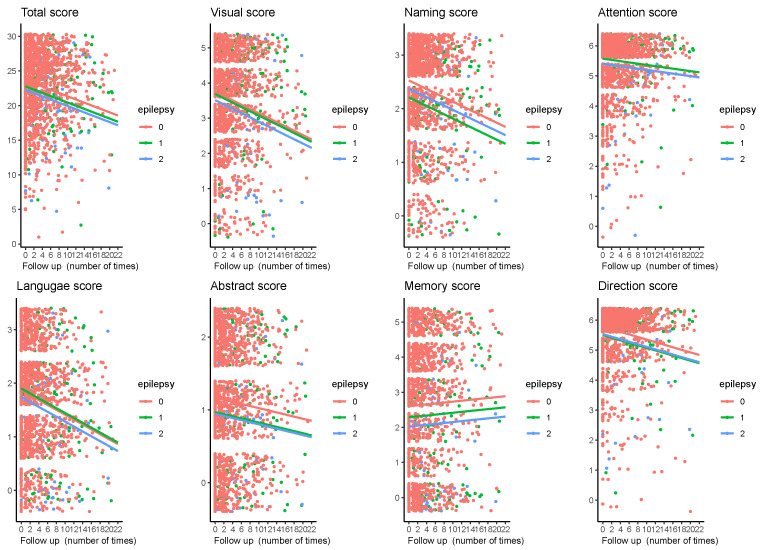
Changes in the MoCA total and subitem scores during follow-up among all patients included in our study. Individual values per visit are shown. The horizontal axis represents the number of follow-up visits (patients were followed up every six months), and the vertical axis represents the score on the MoCA scale. Lines indicate the multivariate-adjusted annual progression rate based on linear mixed-effect modeling.

**Figure 3 cancers-17-01976-f003:**
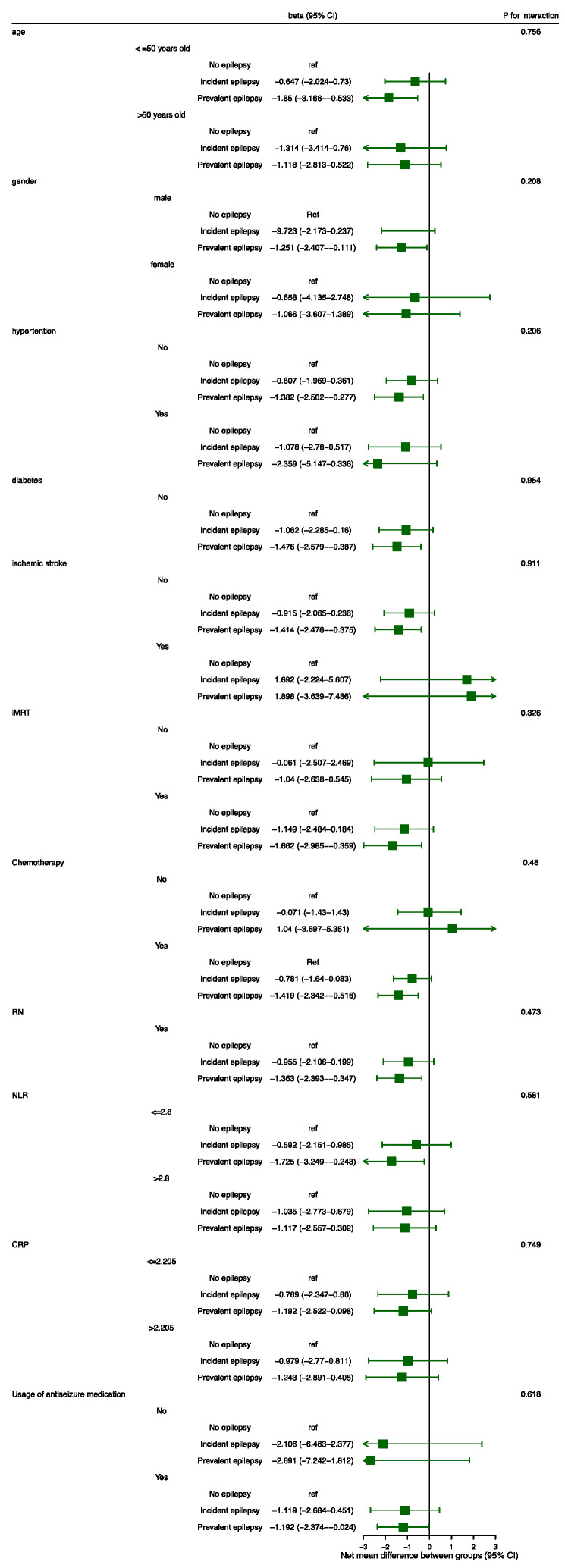
Subgroup analysis. The beta values and their 95%CIs were calculated by means of linear mixed-effect modeling. Abbreviations: CI = confidence interval; CRP = c-reactive protein; IMRT = intensity-modulated radiotherapy; RN = radiation-induced brain necrosis; NLR = neutrophil-to-lymphocyte ratio.

**Table 1 cancers-17-01976-t001:** Baseline characteristics of patients included in our study.

Characteristic	Patients Without Epilepsy at Baseline or During Follow-Up (*n* = 454)	Patients with Incident Epilepsy at Baseline (*n* = 42)	Patients with Prevalent Epilepsy During Follow-Up (*n* = 25)	*p* Value
Demographics				
Age, y, median (IQR)	49.67 (27.28, 72.00)	53.17 (33.95, 66.51)	50.22 (43.53, 59.23)	0.573
BMI, mean (SD)	22.00 (4.27)	22.14 (2.70)	22.29 (4.15)	0.625
Male, *n*, (%)	331 (72.9)	35 (83.3)	17 (68.0)	0.283
Education				0.028
Primary school, *n*, (%)	61 (13.4)	6 (14.3)	4 (16.0)	
Junior middle school, *n*, (%)	126 (27.8)	10 (23.8)	7 (28.0)	
Senior middle school, *n*, (%)	112 (24.7)	10 (23.8)	8 (32.0)	
Undergraduate, *n*, (%)	144 (31.7)	16 (38.1)	6 (24.0)	
Postgraduate, *n*, (%)	11 (2.4)	0 (0.0)	0 (0.0)	
Health behaviors				
Current Smoking, *n*, (%)	18 (4.0)	3 (7.1)	2 (8.0)	0.718
Current Alcohol use, *n*, (%)	7 (1.5)	3 (7.1)	0 (0.0)	0.685
Tumor-related history				
TNM stage, *n*	228	26	18	0.751
Stage I/II, *n*, %	27 (11.8)	4 (15.4)	3 (16.7)	
Stage III/IV, *n*, %	201 (88.2)	22 (84.6)	15 (83.3)	
iMRT, *n*, %	123 (27.1)	5 (11.9)	9 (36.0)	0.054
RT dose (tumor), Gy, median (IQR),	69.0 (42.0, 78.0)	69.0 (69.0, 74.0)	69.0 (68.0, 72.0)	0.081
RT dose (lymph node), Gy, median (IQR)	57.0 (0, 70.0)	59.0 (57.0, 66.0)	57.0 (52.0, 66.0)	0.494
Chemotherapy, *n*, %	203 (44.7)	19 (45.2)	16 (64.0)	0.169
Clinical comorbidity				
Hypertension, *n*, (%)	56 (12.3)	7 (16.7)	3 (12.0)	0.095
Diabetes, *n*, (%)	14 (3.1)	2 (4.8)	4 (16.0)	0.005
Stroke, *n*, (%)	18 (4.0)	3 (7.1)	2 (8.0)	0.718
Radiation-induced brain necrosis, *n*, (%)	400 (88.1)	40 (95.2)	25 (100.0)	0.006
Hypothyroidism, *n*, (%)	49 (10.8)	3 (7.1)	0 (0.0)	0.021
Extracranial arterial stenosis, *n*, %	7 (1.5)	0 (0.0)	0 (0.0)	0.379
Baseline Blood test				
NLR, median (IQR)	2.84 (2.01, 4.45)	2.58 (1.83, 4.45)	2.40 (1.78, 5.73)	0.558
LDL cholesterol, mmol/L, median (IQR)	3.30 (2.71, 3.98)	2.86 (2.44, 3.48)	3.17 (2.50, 3.81)	0.136
HDL cholesterol, mmol/L, median (IQR)	1.25 (1.03, 1.47)	1.20 (1.05, 1.47)	1.36 (1.03, 1.52)	0.629
CRP, mg/L, median (IQR),	2.21 (0.81, 5.97)	1.89 (1.14, 5.81)	2.57 (0.51, 5.63)	0.036
ESR, mm/h, median (IQR),	20.00 (11.75, 38.00)	15.00 (5.00, 22.50)	14.50 (8.25, 26.25)	0.087
Epilepsy parameters				
Latency to seizure, y, median (IQR),	N/A	8.25 (2.07, 14.00)	7.33 (3.02, 11.54))	0.331
ASM use, *n*, %	1 (0.22)	37 (88.10)	12 (48.00)	<0.001
No. of ASMs				<0.001
1, *n*, %	1 (0.22)	22 (52.38)	12 (48.00)	
2, *n*, %	0 (0.00)	14 (33.33)	0 (0.00)	
>2, *n*, %	0 (0.00)	1 (2.38)	0 (0.00)	
Status epilepticus, *n*, %	N/A	1 (2.38)	0 (0.00)	0.437

Abbreviations: NLR = neutrophil-to-lymphocyte ratio; LDL = low-density lipoprotein cholesterol level; HDL = high-density lipoprotein cholesterol level; CRP = c-reactive protein; ESR = erythrocyte sedimentation rate; IMRT = intensity-modulated radiotherapy; RT = radiotherapy. χ^2^ tests were used for the comparison of categorical variables (gender, health behaviors, TNM stage, chemotherapy, clinical comorbidity, ASM use, no. of ASMs, and no. of status epilepticus). ANOVA tests were used for the comparison of continuous variables with a normal distribution (age and BMI). The Mann–Whitney test was used for the comparison of continuous variables with a non-normal distribution (baseline blood test and latency to seizure).

**Table 2 cancers-17-01976-t002:** MoCA total and sub-items scores at baseline among all patients in our study.

	Patients Without Epilepsy (*n* = 454)	Patients with Incident Epilepsy (*n* = 42)	Patients with Prevalent Epilepsy (*n* = 25)	*p* Value
Total score, mean (SD)	22.85 (5.00)	22.11 (5.44)	24.04 (4.04)	0.793
Visual, mean (SD)	3.69 (1.30)	3.69 (1.62)	3.40 (1.35)	0.184
Naming, mean (SD)	2.40 (0.84)	2.41 (0.86)	2.68 (0.56)	0.018
Attention, mean (SD)	5.37 (1.02)	5.41 (1.08)	5.40 (1.15)	0.951
Language, mean (SD)	1.71 (0.94)	1.71 (0.97)	1.96 (0.93)	0.505
Abstract, mean (SD)	0.98 (0.82)	1.14 (0.84)	1.08 (0.76)	0.450
Memory, mean (SD)	2.45 (1.75)	2.19 (1.69)	2.08 (1.75)	0.110
Direction, mean (SD)	5.59 (0.84)	5.60 (0.80)	5.52 (1.01)	0.768
Cognitive function stratification *				
Normal cognitive function, *n*, %	163 (35.90)	15 (35.71)	8 (32.0)	0.525
Mild cognitive dysfunction, *n*, %	222 (48.90)	18 (42.86)	14 (56.0)	
Moderate cognitive dysfunction, *n*, %	63 (13.88)	9 (21.43)	2 (8.0)	
Severe cognitive dysfunction, *n*, %	6 (1.32)	0 (0.0)	1 (4.0)	

* Cognitive dysfunction was categorized into mild cognitive dysfunction (18–25 points), moderate cognitive dysfunction (10–17 points), and severe cognitive dysfunction (<10 points).

**Table 3 cancers-17-01976-t003:** Progression rates every six months of MoCA scores among patients in different groups.

	Univariate Analysis of Progression Rate Every Six Months			Multivariate Analysis of Progression Rate Every Six Months		
	Est	95% CI	*p*	Est	95% CI	*p*
Total score						
No epilepsy	ref			Ref		
Incident epilepsy	−0.28	−0.87, 0.31	0.357	−0.896	−2.026, 0.237	0.126
Prevalent epilepsy	−1.38	−2.18, −0.58	0.001	−1.407	−2.419, −0.412	0.007
Visual						
No epilepsy	ref			Ref		
Incident epilepsy	0.07	−0.090, 0.237	0.376	−0.066	−0.382, 0.250	0.682
Prevalent epilepsy	−0.189	−0.404, 0.026	0.085	−0.240	−0.512, 0.032	0.084
Naming						
No epilepsy	Ref			Ref		
Incident epilepsy	−0.216	−0.332, −0.099	0.000	−0.316	−0.536, −0.094	0.006
Prevalent epilepsy	−0.146	−0.302, 0.010	0.068	−0.154	−0.348, 0.039	0.126
Attention						
No epilepsy	Ref			Ref		
Incident epilepsy	0.027	−0.132, 0.186	0.740	0.163	−0.139, 0.462	0.294
Prevalent epilepsy	−0.251	−0.458, −0.044	0.018	−0.005	−0.261, 0.248	0.967
Language						
No epilepsy	Ref			Ref		
Incident epilepsy	−0.021	−0.151, 0.110	0.757	0.031	−0.238, 0.299	0.823
Prevalent epilepsy	−0.216	−0.386, −0.046	0.013	−0.130	−0.361, 0.101	0.271
Abstract						
No epilepsy	Ref			Ref		
Incident epilepsy	−0.129	−0.387, 0.129	0.329	−0.199	−0.413, 0.015	0.068
Prevalent epilepsy	−0.388	−0.731, −0.048	0.027	−0.224	−0.408, −0.041	0.016
Memory						
No epilepsy	Ref			Ref		
Incident epilepsy	−0.062	−0.297, 0.173	0.604	−0.317	−0.768, 0.134	0.168
Prevalent epilepsy	−0.576	−0.996, −0.156	0.007	−0.581	−0.969, −0.194	0.003
Direction						
No epilepsy	Ref			Ref		
Incident epilepsy	−0.030	−0.166, 0.105	0.663	−0.276	−0.520, −0.033	0.026
Prevalent epilepsy	−0.223	−0.405, −0.042	0.016	−0.243	−0.453, −0.032	0.024

Estimates adjusted for age, sex, education, follow-up time, epilepsy status, history of ischemic stroke, hypertension, diabetes, chemotherapy, iMRT, radiotherapy dose, RN, NLR, CRP, and usage of antiseizure medication. Abbreviations: CRP = c-reactive protein; IMRT = intensity-modulated radiotherapy; RN = radiation-induced brain necrosis; NLR = neutrophil-to-lymphocyte ratio.

**Table 4 cancers-17-01976-t004:** Multivariate analysis for MoCA score changes during follow-up.

Characteristics	Est	95% CI	*p*
Demographics			
Age	−0.033	−0.054, −0.011	0.004
Sex, male vs. female	0.056	−0.349, 0.460	0.788
Education			0.001
<12 years	Ref		
≥12 years	0.303	0.130, 0.479	
Tumor-related history			
Chemotherapy, yes vs. no	−0.257	−0.695, 0.177	0.254
iMRT, yes vs. no	−0.234	−0.722, 0.241	0.339
RT dose (tumor)	0.037	−0.0001, 0.074	0.055
RT dose (lymph node)	7.365 × 10^−5^	−0.014, 0.014	0.992
Clinical comorbidity			
Ischemic stroke, yes vs. no	−0.446	−1.232, 0.331	0.270
Hypertension, yes vs. no	0.620	0.083, 1.152	0.026
Diabetes, yes vs. no	−0.156	−1.137, 0.822	0.758
RN, yes vs. no	−0.015	−0.622, 0.588	0.961
Epileptic status			
No epilepsy	Ref		
Incident epilepsy	−0.896	−2.026, 0.237	0.126
Prevalent epilepsy	−1.407	−2.419, −0.412	0.007
Baseline test			
Baseline MoCA score	0.786	0.744, 0.827	<0.001
NLR	−0.034	−0.101, 0.032	0.331
CRP	−0.005	−0.015, 0.005	0.331
Measurement times	−0.232	−0.294, −0.170	<0.001
Use of antiseizure medication	0.637	−0.466, 1.740	0.265

Abbreviations: CRP = c-reactive protein; iMRT = intensity-modulated radiotherapy; RN = radiation-induced brain necrosis; NLR = neutrophil-to-lymphocyte ratio. RT = radiotherapy.

## Data Availability

The datasets used and/or analyzed during the current study are available from the corresponding author on reasonable request.

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
