# Peer review of "Cognitive Decline in Nasopharyngeal Carcinoma Survivors with Post-Radiation Epilepsy: A Prospective Cohort Study"

_cancers, 2025, doi:10.3390/cancers17121976_

Round 1

Reviewer 1 Report

Comments and Suggestions for Authors

An interesting registry-based cohort study was conducted to investigate cognitive function in nasopharyngeal carcinoma (NPC) patients with epilepsy following radiotherapy. The findings revealed that, at baseline, more than 60% of the patients exhibited mild cognitive impairment. The relevant limitations have also been discussed. Let's have a discussion with the following questions.

  • Besides epilepsy, the neurological complications of cancer present as diffuse (e.g., delirium or dementia), focal (e.g., hemiplegia or aphasia), or multifocal (e.g., left hemiplegia and right visual field defect) disorders. Why did the author merely conduct an in-depth study on epilepsy?
  • Line 38: "Early identification and control of seizure attacks are extremely valuable for mitigating cognitive decline." However, what specific strategies or approaches should be implemented to achieve this goal?
  • Apart from epilepsy caused by treatment or other factors, as a systemic disease, is NPCitself also a major cause of epilepsy? As Line 289 also indicates “Our study showed that cognitive dysfunction is a common problem in NPC patients with epilepsy”, how can a long-term and effective early warning mechanism be established to prevent the occurrence of epilepsy from the source instead of making preparations only after its appearance?
  • In Table 2, both the “education” factor and “diabetes” demonstrate significant associations with epilepsy. How might these relationships be explained?
  • In the 'Introduction' section, it would be advantageous to enhance the new information regarding NPC. A reviewoffers an innovative perspective that conceptualizes NPC as an ecological disease (https://pubmed.ncbi.nlm.nih.gov/37056571/). It is advisable to review this paper for its comprehensive insights into NPC background.

Author Response

An interesting registry-based cohort study was conducted to investigate cognitive function in nasopharyngeal carcinoma (NPC) patients with epilepsy following radiotherapy. The findings revealed that, at baseline, more than 60% of the patients exhibited mild cognitive impairment. The relevant limitations have also been discussed. Let's have a discussion with the following questions.

  1. Besides epilepsy, the neurological complications of cancer present as diffuse (e.g., delirium or dementia), focal (e.g., hemiplegia or aphasia), or multifocal (e.g., left hemiplegia and right visual field defect) disorders. Why did the author merely conduct an in-depth study on epilepsy?

Response: We appreciate the reviewer’s comment. We pay special attention to epilepsy because according to the epidemiological survey data, secondary epilepsy was one of the most common complications in NPC patients after radiotherapy, especially for those with radiation-induced brain necrosis.1 In addition, epilepsy can be well controlled by medication. Thus, proving that controlling seizure attack can slow down the deterioration of cognitive function is extremely valuable and maybe change clinical practice.

  1. Line 38: "Early identification and control of seizure attacks are extremely valuable for mitigating cognitive decline." However, what specific strategies or approaches should be implemented to achieve this goal?

Response: The reviewer has raised a good question about the clinical application of our research. We can achieve early identification and control of seizure attacks from the following aspects. First, strengthen patient education. We should tell patients which symptoms are common manifestation of epilepsy and the risk of seizure attacks. Secondly, we could stratify the risk factors of epilepsy for these patients. For example, in the previous study, we have developed an easily applied nomogram for the prediction of RN-related epilepsy in a large RN cohort.2 It would allow for earlier initiation of medical intervention and provide precautions for patients and their families.

  1. Apart from epilepsy caused by treatment or other factors, as a systemic disease, is NPC itself also a major cause of epilepsy? As Line 289 also indicates “Our study showed that cognitive dysfunction is a common problem in NPC patients with epilepsy”, how can a long-term and effective early warning mechanism be established to prevent the occurrence of epilepsy from the source instead of making preparations only after its appearance?

Response: Thanks for the comments. We think that nasopharyngeal carcinoma itself is not a risk factor for epilepsy. However, radiotherapy for nasopharyngeal carcinoma can lead to brain damage and subsequent recurrent epileptic seizure attacks. In the previous study, we have developed an easily applied nomogram for the prediction of RN-related epilepsy in a large RN cohort.2 We identified seven variables which were significant predictors of epilepsy, including MRI lesion volume, creatine phosphokinase, the maximum radiation dose to the temporal lobe, RN treatment, history of hypertension and/or diabetes, sex, and total cholesterol level. It may be useful in guiding physicians to formulate personalized treatment approaches to prevent the occurrence of epilepsy.

  1. In Table 2, both the “education” factor and “diabetes” demonstrate significant associations with epilepsy. How might these relationships be explained?

Response: We are thankful for the critical question. Table 2 in our manuscript is about MOCA total and sub-items scores at baseline among all patients. Table 1 is about baseline characteristics of patients. We indeed found that more patients in epilepsy group suffered from diabetes than those non-epilepsy. Although we don't know what the specific reason is, the conclusion that diabetes is related to epilepsy has also been reported in the previous literature which may be related to the metabolic disorder and cerebral microvascular disease caused by diabetes, leading to local ischemia or gliosis, and reducing the epilepsy threshold.3

  1. In the 'Introduction' section, it would be advantageous to enhance the new information regarding NPC. A review offers an innovative perspective that conceptualizes NPC as an ecological disease (https://pubmed.ncbi.nlm.nih.gov/37056571/). It is advisable to review this paper for its comprehensive insights into NPC background.

Response: We sincerely appreciate the reviewer's constructive suggestion regarding the conceptualization of nasopharyngeal carcinoma (NPC) as an ecological disease. While the cited review (PMID: 37056571) indeed provides a valuable perspective on NPC pathogenesis, after careful consideration, we have chosen to maintain the current focus of our introduction because our study specifically investigates radiotherapy-induced epilepsy and cognitive dysfunction in NPC survivors, rather than the ecological or etiological aspects of NPC itself. The current introduction already highlights the clinical relevance of late complications in this population. To maintain tight focus on our research question, we have intentionally limited background discussion to factors directly pertaining to radiotherapy sequelae.

We fully acknowledge the importance of multidisciplinary perspectives on NPC, and will certainly consider incorporating ecological aspects in future studies addressing broader NPC pathogenesis.

Reviewer 2 Report

Comments and Suggestions for Authors

Dear authors, your manuscript 

addresses a relevant clinical issue regarding cognitive decline in nasopharyngeal carcinoma (NPC) survivors with post-radiation epilepsy. The research question is significant, methodology robust, and results interesting. However, I would like to suggest a few areas that require improvements: 

Lines 55-60, you mentioned "vascular risk factors" (VRFs), yet specific examples and their relevance to NPC and epilepsy should be expanded for clarity

Lines 75-85, details about radiation dose and fractionation are missing. These specifics significantly impact cognitive outcomes and could be included.

Line 150, the mean age at study entry is given, but the range or interquartile range would add important context.

Lines 172-200, statistical power or effect size considerations were not mentioned; these are necessary to interpret clinical relevance.

Lines 285-287, advances in recent treatments were mentioned but not detailed; specifying examples would strengthen the discussion.

Thank you very much

Author Response

Dear authors, your manuscript addresses a relevant clinical issue regarding cognitive decline in nasopharyngeal carcinoma (NPC) survivors with post-radiation epilepsy. The research question is significant, methodology robust, and results interesting. However, I would like to suggest a few areas that require improvements: 

  1. Lines 55-60, you mentioned "vascular risk factors" (VRFs), yet specific examples and their relevance to NPC and epilepsy should be expanded for clarity.

Response:We appreciate the reviewer’s comment. Vascular risk factors here refer to risk factors related to atherosclerosis, such as hypertension, diabetes, hyperlipidemia and history cerebrovascular disease. We have clarified it in the revised manuscript in lines 55-59:” For NPC survivors with epilepsy, the cognitive impairment is attributable to a combination of risk factors, including age, treatment strategy for cancer, RN, recurrent seizure attack and vascular risk factors (VRFs, for example hypertension, diabetes, hyperlipidemia and history of cerebrovascular disease).” We play attention to these factors because aggressive targeting of vascular risk factors might affect the integrity of brain networks, which help to prevent seizure.

  1. Lines 75-85, details about radiation dose and fractionation are missing. These specifics significantly impact cognitive outcomes and could be included.

Response: Thanks for the insightful suggestion. In the revised manuscript, we made every effort to supplement the data of the radiation dose, including the radiation dose of the primary lesion and lymph nodes. Both of them were added in the multivariable analysis. However, because most patients in our cohort receive cancer treatment in other hospitals, and there is usually a latency period of several years from the end of radiotherapy to the onset of neurological symptoms, It is difficult to obtain the detailed radiotherapy data for all the patients. Thus, fractionation is still missing in the revised manuscript. We have added them in the limitation.

  1. Line 150, the mean age at study entry is given, but the range or interquartile range would add important context.

Response: Thanks for the insightful suggestion. We have revised the sentence to “Median age at study entry was 49.9 years (IQR, 26.1, 73.4) and 26.4% of participants were women.”

  1. Lines 172-200, statistical power or effect size considerations were not mentioned; these are necessary to interpret clinical relevance.

Response: We appreciate the reviewer’s important comment regarding statistical power and clinical interpretation. Our study was designed as an exploratory observational cohort to characterize associations between epilepsy and dementia in NPC patients, rather than to test a specific hypothesis. Thus, we did not perform a priori sample size calculations. To address clinical relevance, we have reported all effect estimates with 95% confidence intervals to allow readers to evaluate both statistical precision and clinical significance.

Our study was based on a prospectively ongoing cohort study, recruiting head and neck adult cancer patients with radiotherapy-related nervous system complications. To the best of our knowledge, this represents the largest available dataset addressing this specific population, and we have included all eligible cases to maximize the robustness of our observations.

  1. Lines 285-287, advances in recent treatments were mentioned but not detailed; specifying examples would strengthen the discussion.

Response: We thank the reviewer for these comments. we have revised the sentence.

Reviewer 3 Report

Comments and Suggestions for Authors

definition and handling of “incident” versus “prevalent” epilepsy are unclear. It should be defined precisely. 

The similarity index of paper is too high. It needs a comprehensive revision accordingly. 

There is a potential selection bias and low sample size in the prevalent-epilepsy subgroup; discuss statistical power and the possibility that more severe cases were preferentially enrolled. How the sample size was calculated?

missing dosimetric data and anti-seizure medication details may confound results. These information should be provided and the results should be made accordingly.

The use of statistical methods is mentioned not clear enough, Please elaborate on how the tests have been chosen and controlled for which variables. 

abstract and introduction contain long or fragmented sentences and redundant phrases; edit for clarity and concision. The paper needs revision.

table 1 should report exact p-values to three decimals, indicate the statistical test used for each variable, and present effect sizes when significant differences are small.

phrase “normal cognitive dysfunction” (line 170) is contradictory.

typographical issues include unnecessary superscript after “npc” (line 44), split word “sta- tistical” (line 137), and missing plural “scores” in line 181.

Author Response

  1. definition and handling of “incident” versus “prevalent” epilepsy are unclear. It should be defined precisely. 

Response: Our research aimed to investigate the relationship between seizure frequency and cognitive function. We hypothesized that NPC survivors with active epilepsy would experience accelerated cognitive deterioration compared to those without epilepsy. Accordingly, we defined incident epilepsy cases as patients whose frequency of seizure attack was less than twice per year after enrollment into our study. The remaining epilepsy cases were regarded as prevalent epilepsy cases. The control group consisted of participants without epilepsy. Of course, this is not a universal definition, but rather a classification of our research.

  1. The similarity index of paper is too high. It needs a comprehensive revision accordingly. 

Response: We sincerely appreciate the reviewer’s comment regarding the similarity index of our manuscript. We acknowledge that certain sections, particularly in the methods, may have shared phrasing with previously published literature due to standard terminology and methodological descriptions. In the revised manuscript, we have made necessary modifications to the sentences.

  1. There is a potential selection bias and low sample size in the prevalent-epilepsy subgroup; discuss statistical power and the possibility that more severe cases were preferentially enrolled. How the sample size was calculated?

Response: We appreciate the reviewer’s important comment regarding statistical power and clinical interpretation. Our study was designed as an exploratory observational cohort to characterize associations between epilepsy and dementia in NPC patients, rather than to test a specific hypothesis. Thus, we did not perform a priori sample size calculations. To address clinical relevance, we have reported all effect estimates with 95% confidence intervals to allow readers to evaluate both statistical precision and clinical significance. We have list it as a limitation in the revised manuscript.

Our study was based on a prospectively ongoing cohort study, recruiting head and neck adult cancer patients with radiotherapy-related nervous system complications. To the best of our knowledge, this represents the largest available dataset addressing this specific population, and we have included all eligible cases to maximize the robustness of our observations.

  1. missing dosimetric data and anti-seizure medication details may confound results. These information should be provided and the results should be made accordingly.

Response: In the revised manuscript, we have provided anti-seizure medication (ASM) details. Considering anti-seizure drugs were variable among patients, we put usage of ASM or not as a variable in the multivariable analysis, rather than the specific name of drug.

  1. The use of statistical methods is mentioned not clear enough, Please elaborate on how the tests have been chosen and controlled for which variables. 

Response: We sincerely appreciate the reviewer’s insightful comment regarding the clarification of statistical methods. We have provided a detail elaboration on our statistical approach in the revised manuscript in lines 126-146.

  1. abstract and introduction contain long or fragmented sentences and redundant phrases; edit for clarity and concision. The paper needs revision.

Response: we sincerely appreciate the reviewer’s valuable suggestions to improve the clarity and conciseness of our manuscript. We have thoroughly revised the paper.

  1. table 1 should report exact p-values to three decimals, indicate the statistical test used for each variable, and present effect sizes when significant differences are small.

Response:The p-values in table 1 (revised) is reported to three decimals. We also have added statistical test for each variable below the table in lines 174-178: “χ2 tests was used for comparison of categorical variables (gender, health behaviors, TNM stage, chemotherapy, clinical comorbidity). ANOVA tests was used for comparison of continuous variables with a normal distribution (age and BMI). Mann-Whitney test was used for comparison of continuous variables with non-normal distribution (baseline blood test).”

  1. phrase “normal cognitive dysfunction” (line 170) is contradictory.

Response: We have revised the terms to “normal cognitive function” in table 2 (revised version).

  1. typographical issues include unnecessary superscript after “npc” (line 44), split word “sta- tistical” (line 137), and missing plural “scores” in line 181.

Response: We have revised the terms as suggestions.

Reviewer 4 Report

Comments and Suggestions for Authors

Comments:

This is a study evaluating the cognitive decline in nasopharyngeal carcinoma survivors with post radiation epilepsy. In this prospective cohort study, the investigators have included 454 patients without epilepsy at baseline and 42 patients with incident epilepsy at baseline and report rate of decline in MoCA was significantly faster in patients with prevalent epilepsy.  The study design, analysis and their interpretations are appropriate. The manuscript is logically structured and easy to read. The authors need to address the following points:

  • The major concern in the study is the hypothesis. Cognitive decline is multifactorial in such patients. Considering only seizure without it’s details (as I have mentioned later) may not appropriate.
  • MoCA is a screening tool for cognitive dysfunction. For cognitive assessment other elaborative tools could have been better.
  • The definition of “incident epilepsy’ and ‘Prevalent epilepsy” needs to be properly defined with references.
  • Personnel involved in administration of MoCA should be mentioned. Were they aware of the clinical parameters or seizure status of the patient?
  • Figure 1: The Exclusion of 273 does not add up (268+17)
  • Results: Total radiation dose should be mentioned in the baseline Table 1.
  • Results: Baseline information should mention the brain neuroimaging and EEG parameters.
  • Results: Relevant clinical information e.g Onset of seizure after radiotherapy (time/days), presence/absence of status epilepticus, number of ASM in use, seizure frequency, site of radiation necrosis, should be provided. These covariates can influence the cognitive parameters and are essential to be considered in regression analysis.
  • Onset of seizure after radiotherapy is of importance, as early onset seizure and late onset seizures have a different pathophysiology and can affect the cognition differently.
  • Results: The study included patients with cognitive assessment from January 2005 till December 2023. I am not sure how the follow up duration has been shown to be 22 years? (Figure 3)
  • Of the 67 patients with epilepsy, 22 had focal and 45 had generalized seizure. The types of seizure are always not exclusive. Authors may recheck the values.
  • Important co-variates e.g depression, anxiety and other psychiatric co-morbidities which are commonly seen in patients with malignancy can impact cognitive functions. They have not been assessed or addressed in the study.

Author Response

This is a study evaluating the cognitive decline in nasopharyngeal carcinoma survivors with post radiation epilepsy. In this prospective cohort study, the investigators have included 454 patients without epilepsy at baseline and 42 patients with incident epilepsy at baseline and report rate of decline in MoCA was significantly faster in patients with prevalent epilepsy.  The study design, analysis and their interpretations are appropriate. The manuscript is logically structured and easy to read. The authors need to address the following points:

  1. The major concern in the study is the hypothesis. Cognitive decline is multifactorial in such patients. Considering only seizure without it’s details (as I have mentioned later) may not appropriate.

Response: We appreciate the reviewer’s comment. We completely agree that cognitive decline in these patients is multifactorial. In the revised manuscript, we have added the latency period of epilepsy and the information on the use of antiseizure medication (ASM) in the baseline description. while in the multivariate analysis, we also added the variable of whether ASM were used.

  1. MoCA is a screening tool for cognitive dysfunction. For cognitive assessment other elaborative tools could have been better.

Response: We agree that the diagnosis and differential diagnosis of dementia in clinical practice is not solely based on a single MOCA scale, but also involves a combination of patient medical history, physical signs, imaging examinations, and scale assessments. We chose the MoCA as the main tool in our cohort study because it is one of the most widely used cognitive tests and has been shown to have high sensitivity (90-93%) and specificity (85-100%) for mild cognitive impairment (MCI) detection 4,5. Admittedly, this method is not without its limitations, which are acknowledged in our revised manuscript.

  1. The definition of “incident epilepsy’ and ‘Prevalent epilepsy” needs to be properly defined with references.

Response: Our research aimed to investigate the relationship between seizure frequency and cognitive function. We hypothesized that NPC survivors with active epilepsy would experience accelerated cognitive deterioration compared to those without epilepsy. Accordingly, we defined incident epilepsy cases as patients whose frequency of seizure attack was less than twice per year after enrollment into our study. The remaining epilepsy cases were regarded as prevalent epilepsy cases. The control group consisted of participants without epilepsy. of course, this is not a universal definition, but rather a classification of our research.

  1. Personnel involved in administration of MoCA should be mentioned. Were they aware of the clinical parameters or seizure status of the patient?

Response: Assessment of cognition were done through face-to-face interviews by trained interviewers that have the certification of cognition assessment and blind to the seizure status.

  1. Figure 1: The Exclusion of 273 does not add up (268+17)

Response: Sorry for the calculation error. We have corrected the numbers in the revised figure 1. The exclusion should be 285.

  1. Results: Total radiation dose should be mentioned in the baseline Table 1.

Response: We have added radiation dose to the primary lesion and radiation dose to the lymph nodes in the revised table 1.

  1. Results: Baseline information should mention the brain neuroimaging and EEG parameters.

Response: We sincerely appreciate the reviewer’s insightful suggestion regarding the inclusion of brain neuroimaging and EEG parameters, which would undoubtedly provide valuable neurophysiological evidence to complement our findings. However, due to the limited resources, these data were not incorporated in the present study. To ensure transparency, we have now explicitly addressed this limitation in the revised manuscript and highlighted the importance of future investigations incorporating neural-level data.

  1. Results: Relevant clinical information e.g Onset of seizure after radiotherapy (time/days), presence/absence of status epilepticus, number of ASM in use, seizure frequency, site of radiation necrosis, should be provided. These covariates can influence the cognitive parameters and are essential to be considered in regression analysis. Onset of seizure after radiotherapy is of importance, as early onset seizure and late onset seizures have a different pathophysiology and can affect the cognition differently.

Response: We sincerely appreciate the reviewer’s insightful comment. In the revised manuscript, latency of seizure after radiotherapy, use of ASM, presence of status epilepticus were added.

  1. Results: The study included patients with cognitive assessment from January 2005 till December 2023. I am not sure how the follow up duration has been shown to be 22 years? (Figure 2)。

Response: We appreciate the reviewer’s comment. We apology for the mistake in figure 2. Patients in our study were followed up every six months, and the median follow-up period of 3.96 years (iqr 2.0 to 7.74 years), the longest follow-up period is 10.58 years. In figure 2, the horizontal axis represents the number of follow-up visits (patients were followed up every six months), and the vertical axis represents the score of the MoCA scale. Therefore, 22 represents not years, but the number of follow-up visits. Revised figure 2 have been upload.  

  1. Of the 67 patients with epilepsy, 22 had focal and 45 had generalized seizure. The types of seizure are always not exclusive. Authors may recheck the values.

Response: We appreciate the reviewer’s comment. According to the ILAE classification, the diagnosis of epilepsy presents three levels, starting with seizure type, after diagnosis of the seizure type, the next step is diagnosis of epilepsy type, including focal epilepsy, generalized epilepsy, combined generalized, and focal epilepsy, and also an unknown epilepsy group. What we want to express here is the type of epilepsy, not the type of seizure. We apology for the incorrect expression. We have changed the sentence in line 152-152 to “Of the 67 patients with epilepsy, twenty-two were focal epilepsy and forty-five were generalized epilepsy.”

  1. Important co-variates e.g depression, anxiety and other psychiatric co-morbidities which are commonly seen in patients with malignancy can impact cognitive functions. They have not been assessed or addressed in the study.

Response: We appreciate the reviewer’s comment. The evaluation data of emotional scales (such as depression, anxiety) was missing in some of patients in our study, thus we did not include the data. We have explicitly addressed this limitation in the revised manuscript in lines 307-309.

Reviewer 5 Report

Comments and Suggestions for Authors

The manuscript addresses an important clinical issue which is cognitive decline in NPC survivors with post-radiation epilepsy.

However, this manuscript has many caveats and it’s not ready for publication now.

I believe this has potential and can be improved with major revisions. To help the authors improve their work, I provide detailed comments and constructive feedback for the authors.

  1. The abstract reports estimates (e.g., "Est -1.260") without explaining what "Est" refers to (presumably "estimate"). Clarify terms for readability.
  2. The abstract doesn’t mention sample size or follow-up duration in the results.
  3. The abstract mentiones "early identification and control of seizure attack is extremely valuable" but the study design can’t really conclude this.
  4. The introduction doesn’t cite recent studies on cognitive decline in epilepsy (e.g., 2023–2024 papers).
  5. The hypothesis ("NPC survivors with active epilepsy experience accelerated cognitive decline") is not clear. Define "active epilepsy" (e.g., seizure frequency, AED use).
  6. Some methodological details (e.g., MoCA) appear here which should be included in the Methods section. Move MoCA details to the Methods section.
  1. "Incident" vs. "prevalent" epilepsy definitions are arbitrary (e.g., "twice per year" cutoff lacks justification). Cite ILAE guidelines for consistency.
  2. There’s no discussion of handling missing covariate data (e.g., CRP, NLR) in the Methods section.
  3. Important confounders (e.g., radiation dose, volume) are not included.
  4. Add radiation dose/volume as covariates or acknowledge this limitation.
  1. There’s no mention of inter-rater reliability or validation in NPC populations.
  2. Justify epilepsy classification with references to clinical standards.
  1. Table 1 shows significant differences in diabetes, RN, and CRP across groups, but the analysis doesn’t address how these affect outcomes.
  2. Figure 3 is unreadable (labels overlap, no clear beta values).
  3. Multivariable model includes many covariates (e.g., TNM stage, NLR) without justification for inclusion.
  4. Discuss baseline imbalances and their potential impact (e.g., propensity score matching).
  1. The discussion exaggerates implications (e.g., "seizure control can reverse cognitive impairment") without any RCT evidence.
  2. Subgroup analysis showed no significant interactions, but this is not critically discussed.
  3. The limitations section is weak and doesn’t mention key weaknesses (e.g., arbitrary epilepsy classification, missing radiation data).
  4. Expand limitations to include epilepsy definition and unmeasured confounders.
  5. Conclusion is very weak. It repeats the results without adding clinical or research implications.

Author Response

The manuscript addresses an important clinical issue which is cognitive decline in NPC survivors with post-radiation epilepsy.

However, this manuscript has many caveats and it’s not ready for publication now.

I believe this has potential and can be improved with major revisions. To help the authors improve their work, I provide detailed comments and constructive feedback for the authors.

  1. The abstract reports estimates (e.g., "Est -1.260") without explaining what "Est" refers to (presumably "estimate"). Clarify terms for readability.

Response: We have changed the sentence to “The rate of decline in MoCA was significantly faster in patients with prevalent epilepsy compared with no epilepsy after adjusting for demographics, health behaviors, tumor related history, complication, and inflammatory blood index (estimate -1.407, 95%CI -2.419, -0.412, p=0.007).”

  1. The abstract doesn’t mention sample size or follow-up duration in the results.

Response: Thank you for the suggestion. We have added this information in the abstract in line 29-30: “538 patients with a median follow-up period of 3.96 years were include in our study” .

  1. The abstract mentiones "early identification and control of seizure attack is extremely valuable" but the study design can’t really conclude this.

Response: We appreciate the reviewer’s comment. We have changed the sentence to “Control of seizure attack may be valuable to mitigate cognitive decline”.

  1. The introduction doesn’t cite recent studies on cognitive decline in epilepsy (e.g., 2023–2024 papers).

Response: We sincerely appreciate your insightful comment regarding the inclusion of recent literature. In the revised manuscript, we have added citations from the recent studies on cognitive decline in epilepsy (reference 10,11). These additions strengthen our introduction’s contextual framework and are highlighted in the revised manuscript.

  1. The hypothesis ("NPC survivors with active epilepsy experience accelerated cognitive decline") is not clear. Define "active epilepsy" (e.g., seizure frequency, AED use).

Response:We sincerely appreciate the reviewer’s comment. In the revised manuscript, we have clarified the definition of “active epilepsy” in lines 99-102: “Active epilepsy means having experienced at least one unprovoked seizure in the preceding 12 months, regardless of whether they were taking ASM or not. To investigate the impact of seizure control on cognitive function, we identified seizures as a major variable”.  

  1. Some methodological details (e.g., MoCA) appear here which should be included in the Methods section. Move MoCA details to the Methods section.

Response: MoCA details is shown in the Methods section in line 107-115.

  1. "Incident" vs. "prevalent" epilepsy definitions are arbitrary (e.g., "twice per year" cutoff lacks justification). Cite ILAE guidelines for consistency.

Response: Our research aimed to investigate the relationship between seizure frequency and cognitive function. We hypothesized that NPC survivors with active epilepsy would experience accelerated cognitive deterioration compared to those without epilepsy. Accordingly, we defined incident epilepsy cases as patients whose frequency of seizure attack was less than twice per year after enrollment into our study. The remaining epilepsy cases were regarded as prevalent epilepsy cases. The control group consisted of participants without epilepsy. of course, this is not a universal definition, but rather a classification of our research.

  1. There’s no discussion of handling missing covariate data (e.g., CRP, NLR) in the Methods section.

Response: Thank you for the comment. In our study, we did not perform imputation or adjustment for the missing data. The proportion of missing data for these covariates was very low. 

  1. Important confounders (e.g., radiation dose, volume) are not included. Add radiation dose/volume as covariates or acknowledge this limitation.

Response: Thanks for the insightful suggestion. In the revised manuscript, we made every effort to supplement the data of the radiation dose, including the radiation dose of the primary lesion and lymph nodes. Both of them were added in the multivariable analysis. However, because most patients in our cohort receive cancer treatment in other hospitals, and there is usually a latency period of several years from the end of radiotherapy to the onset of neurological symptoms, it is difficult to obtain the detailed radiotherapy data for all the patients. Thus, fractionation is still missing in the revised manuscript. We have added them in the limitation.

  1. There’s no mention of inter-rater reliability or validation in NPC populations.

Response:We thank the reviewer for raising this important point. In our study, assessment of cognition was done through face-to-face interviews by trained interviewers that have the certification of cognition assessment and blind to the seizure status. However, we acknowledge that inter-rater reliability (IRR) was not formally quantified, which is a limitation.

  1. Justify epilepsy classification with references to clinical standards.

Response: We have added references in the revised manuscript.

  1. Table 1 shows significant differences in diabetes, RN, and CRP across groups, but the analysis doesn’t address how these affect outcomes.

Response: We thank the reviewer for this comment. In table 1, education, diabetes, RN, hypothyroidism, CRP level were not balance among the three group. We have put them into the multivariate analysis. The reasons about baseline imbalance was also been discussed in the discussion section.

  1. Figure 3 is unreadable (labels overlap, no clear beta values).

Response: We have redrawn figure 3.

  1. Multivariable model includes many covariates (e.g., TNM stage, NLR) without justification for inclusion.

Response:We sincerely appreciate the reviewer's valuable comment regarding the selection of covariates in our multivariable analysis. The covariates included in our multivariable model were selected based on the univariate analysis. Also Clinical relevance as potential confounders of cognition were included in multivariable model, although some of these variables did not show statistical differences in univariate analysis. Hypothyroidism was not included in the model because the case was too small. We have added explicit covariate selection criteria to Methods in line 137-141: “Variables with P value < 0.1 in Univariate analysis or of clinical interest (age, gender, education, intensity-modulated radiation therapy, radiation dose, hypertension, diabetes mellitus, stroke, usage of ASM, RN, NLR and CRP) were further evaluated in the multivariable linear mixed model. ”

  1. Discuss baseline imbalances and their potential impact (e.g., propensity score matching).

Response:We thank the reviewer for highlighting this important issue. In the baseline, education, diabetes, RN, hypothyroidism, CRP level were not balance among the three group. Although we did not perform propensity score matching, we have put them into the multivariate analysis.

  1. The discussion exaggerates implications (e.g., "seizure control can reverse cognitive impairment") without any RCT evidence.

Response:We must admit that there is no evidence of RCT for this statement.

  1. Subgroup analysis showed no significant interactions, but this is not critically discussed.

Response: We have added some discussion about the subgroup analysis.

  1. The limitations section is weak and doesn’t mention key weaknesses (e.g., arbitrary epilepsy classification, missing radiation data).Expand limitations to include epilepsy definition and unmeasured confounders.

Response: We have made revisions to limitation in lines 302-321: “There are several important limitations of our study. First, although it is as prospective study, some patients may lose to follow-up because of disease progression or unexpected death, which could lead to potential biases. Second, some detail but important information on radiotherapy for example radiation fractionation was not available. In order to minimize the confounding factors of radiation strategy, we only included patients with NPC. Besides, we did not have data on brain neuroimaging and EEG parameters at baseline, as well as other psychiatric comorbidities (such as depression, anxiety) which are commonly seen in patients with malignancy can also impact cognitive functions. Third, part of our participants had concurrent RN, and may receive corticosteroid or bevacizumab therapy6,7. The changing volume of RN during the follow-up might also partially affect the cognition evaluation. Fourth, we used data collected from 2005 to 2023 in order to enroll as many patients as possible. Despite the cohort is large, the number of patients with epilepsy was still small (only 67 patients), likely limiting the power. Fifth, MoCA is only a screening tool for cognitive dysfunction. The diagnosis and severity of dementia involve a combination of patient medical history, physical signs, imaging examinations, and scale assessments. In the future, using more elaborative tools may help to evaluate the cognitive function better. Finally, change in epilepsy treatment (drugs and surgery), seizure frequency and RN treatment (corticosteroid and bevacizumab) strategies during follow-up may also have important implications for cognitive health, which have not been fully evaluated in our study.”

  1. Conclusion is very weak. It repeats the results without adding clinical or research implications.

Response: We agree with the reviewer that our original conclusion underemphasized the implications. We have thoroughly revised the Conclusion section in lines 323-329: “Our study showed that cognitive dysfunction is a common problem in NPC patients with epilepsy. Furthermore, global cognitive function declined more rapidly in NPC patients with prevalent epilepsy compared with those without epilepsy. Reducing the frequency of seizure attacks and treating hypertension may be modifiable factors in slow cognitive decline. Further studies to founding the biological mechanisms to explain the association and potential identify more modifiable risk factors for slowing down the cognitive deterioration are warranted. “

Round 2

Reviewer 3 Report

Comments and Suggestions for Authors

The paper is improved. thanks

Author Response

Thank you.

Reviewer 4 Report

Comments and Suggestions for Authors

Comments:

The authors have submitted their replies to my previous queries. Most points have been addressed. However, some corrections are still required. They are:

  • In abstract, the no of patients has been mentioned as 538. This is not correct. The flow chart mentions 521 patients.
  • The RT dose (In table 1) should be provided with their units.
  • The median latency to seizure after radiotherapy along with it’s range should be provided for both ’incident epilepsy’ and ‘prevalent epilepsy’ in Table 1.
  • The epilepsy parameters e.g ASM use, no of ASM, status etc should be provided in Table 1 for each groups.
  • The % of women would be 27.09%, not 26.4%. (In text of results section)

Author Response

Comments:

The authors have submitted their replies to my previous queries. Most points have been addressed. However, some corrections are still required. They are:

  1. In abstract, the no of patients has been mentioned as 538. This is not correct. The flow chart mentions 521 patients.

Response: We apologized for the mistake. We have revised the sentence in line 29 to “521 patients with a median follow-up period of 3.96 years were included in our study.”

  1. The RT dose (In table 1) should be provided with their units.

Response: The units of RT dose (Gy) has been provided in table 1.

  1. The median latency to seizure after radiotherapy along with it’s range should be provided for both ’incident epilepsy’ and ‘prevalent epilepsy’ in Table 1.

Response: We appreciate the reviewer’s comment. The median latency with range to seizure after radiotherapy have been provided in table 1.

  1. The epilepsy parameters e.g ASM use, no of ASM, status etc should be provided in Table 1 for each groups.

Response:we sincerely appreciate the reviewer’s valuable suggestions. ASM use, no of ASM, status epilepticus have been added in Table 1 for each groups.

  1. The % of women would be 27.09%, not 26.4%. (In text of results section)

Response: The % of women was 26.5% (138/521). We have revised the sentence in line 159-160 to : “Median age at study entry was 49.9 years (IQR, [26.1, 73.4])and 26.5% (138/521) of participants were women.”

Reviewer 5 Report

Comments and Suggestions for Authors

The authors have addressed my comments well.

Author Response

Thank you.